

# Sorted gene genealogies and species-specific nonsynonymous substitutions point to putative postmating prezygotic isolation genes in *Allonemobius* crickets

Suegene Noh[1] and Jeremy L. Marshall[2]

[1] Department of Biology, Washington University in St. Louis, St. Louis, MO, United States
[2] Department of Entomology, Kansas State University, Manhattan, KS, United States

## ABSTRACT

In the *Allonemobius socius* complex of crickets, reproductive isolation is primarily accomplished via postmating prezygotic barriers. We tested seven protein-coding genes expressed in the male ejaculate for patterns of evolution consistent with a putative role as postmating prezygotic isolation genes. Our recently diverged species generally lacked sequence variation. As a result, $\omega$-based tests were only mildly successful. Some of our genes showed evidence of elevated $\omega$ values on the internal branches of gene trees. In a couple of genes, these internal branches coincided with both species branching events of the species tree, between *A. fasciatus* and the other two species, and between *A. socius* and *A. sp. nov.* Tex. In comparison, more successful approaches were those that took advantage of the varying degrees of lineage sorting and allele sharing among our young species. These approaches were particularly powerful within the contact zone. Among the genes we tested we found genes with genealogies that indicated relatively advanced degrees of lineage sorting across both allopatric and contact zone alleles. Within a contact zone between two members of the species complex, only a subset of genes maintained allelic segregation despite evidence of ongoing gene flow in other genes. The overlap in these analyses was *arginine kinase* (AK) and *apolipoprotein A-1 binding protein* (APBP). These genes represent two of the first examples of sperm maturation, capacitation, and motility proteins with fixed non-synonymous substitutions between species-specific alleles that may lead to postmating prezygotic isolation. Both genes express ejaculate proteins transferred to females during copulation and were previously identified through comparative proteomics. We discuss the potential function of these genes in the context of the specific postmating prezygotic isolation phenotype among our species, namely conspecific sperm precedence and the superior ability of conspecific males to induce oviposition in females.

Corresponding author
Suegene Noh,
suegene.noh@gmail.com

## INTRODUCTION

Not all genes contribute equally to reproductive isolation during speciation. 'Speciation' (*Wu, 2001*; *Wu & Ting, 2004*; *Nosil & Schluter, 2011*), 'isolation' (*Rieseberg, Church & Morjan, 2004*), or 'barrier' (*Noor & Feder, 2006*) genes are expected to show very different

patterns of evolution compared to genes that are not directly involved in reproductive isolation when species are still undergoing lineage sorting (*Wu, 2001*; *Feder, Egan & Nosil, 2012*). Therefore, we expect to find putative speciation genes among those genes that become fixed for alternative alleles within each incipient species early in the process of divergence, with said alleles rarely crossing the species boundary in sympatry (*Ting, Tsaur & Wu, 2000*; *Dopman et al., 2005*).

Rapidly evolving reproductive proteins that can affect fertilization success have an important role in the evolution of postmating prezygotic reproductive isolation. Many reproductive genes are known to evolve rapidly in a variety of organisms (*Civetta & Singh, 1998*; *Swanson & Vacquier, 2002*; *Clark, Aagaard & Swanson, 2006*; *Panhuis & Swanson, 2006*; *Snook et al., 2009*). In *Drosophila*, where some of the most extensive work has been done, genes that show male-biased expression evolve faster compared to female-biased and somatically expressed genes (*Zhang, Hambuch & Parsch, 2004*; *Zhang & Parsch, 2005*; *Metta et al., 2006*; *Pröschel, Zhang & Parsch, 2006*; *Haerty et al., 2007*). Seminal fluid proteins in particular tend to show an excess of non-synonymous substitutions (*Begun et al., 2000*; *Swanson et al., 2001*; *Wagstaff & Begun, 2005*; *Almeida & DeSalle, 2008*). Similar patterns have also been observed in mice and primates (*Clark & Swanson, 2005*; *Karn et al., 2008*; *Ramm et al., 2008*; *Turner, Chuong & Hoekstra, 2008*; *Dean et al., 2009*). Because they enable us to isolate male reproductive protein-coding genes that can directly interact with their female counterparts, proteomic analyses of insect spermatophores have proved to be particularly effective for narrowing prospects in the search for putative speciation genes (*Andrés, Maroja & Harrison, 2008*; *Marshall et al., 2011*; *Andrés et al., 2013*).

The male ejaculate proteome comprises sperm-expressed proteins and seminal fluid proteins. Sperm not only contribute half of the diploid genome, but are also involved in sperm-egg interactions, including egg activation and the delivery of paternal factors during fertilization (*Dorus et al., 2006*). The majority of seminarl fluid proteins are produced by male accessory glands. These proteins contain conserved functional classes of peptides and pro-hormones that are involved in sperm binding, proteolysis, lipid metabolism, and immune function (*Mueller et al., 2004*; *Chapman & Davies, 2004*; *Poiani, 2006*; *Avila et al., 2011*). Once transferred into the female reproductive tract, these proteins can initiate a wide-range of physiological functions, including increased egg production and oviposition, decreased receptivity, decreased lifespan, and increased feeding in females (reviewed in *Avila et al., 2011*). The interacting female counterparts to these ejaculate proteins (EPs) are not as well known (*Ram, Ji & Wolfner, 2005*; *Ram & Wolfner, 2007*; *Snook et al., 2009*) though genomic data is proving to be invaluable for identifying candidates (*Baer et al., 2009a*; *Findlay et al., 2014*). The evolution of EPs has been hypothesized to be driven by various processes, including female sperm preference, sperm competition, and sexual conflict (*Mueller et al., 2004*; *Snook et al., 2009*).

The *Allonemobius socius* complex of ground crickets, *A. socius*, *A. fasciatus*, and *A. sp. nov.* Tex, represents a powerful system to explore the hypothesized link between EP divergence and reproductive isolation. Members of this complex are primarily isolated from one another by two postmating, prezygotic phenotypes—conspecific sperm precedence (*Gregory & Howard, 1994*; *Howard et al., 1998a*; *Howard et al., 1998b*; *Marshall, 2004*)

and the superior ability of conspecific males to induce females to lay eggs (*Gregory & Howard, 1993*; *Howard et al., 1998b*). Two other compelling features of this study system are species boundaries that remain intact in sympatry despite some gene flow (*Howard, 1986*; *Howard & Waring, 1991*; *Traylor et al., 2008*) and the very recent nature of divergence between species (i.e., within the last 30,000 years; *Marshall, 2004*; *Marshall, 2007*). Indeed, divergence is so recent that few species-specific alleles have been identified. Only 2 of 17 allozyme markers (*Howard, 1983*), 2 of 5,400 AFLP markers (*Howard et al., 2002*), ∼21 of 1,660 thorax proteins and ∼33 of 922 ejaculate proteins (*Marshall et al., 2011*), and 1 of 16 randomly chosen reproductive genes spanning >7,500 bp of coding sequence (J Marshall, 2016, unpublished data) yield evidence of species specificity. Taken together, the above data suggest that while there is sufficient genetic divergence to produce reproductive isolation and maintain species boundaries in sympatry, the vast majority of genes show little evidence of divergence. In all, the *A. socius* complex represents a system whereby speciation is ongoing with potentially relatively few genes contributing to postmating prezygotic reproductive isolation between species. Therefore, if we can identify those ejaculate and female reproductive tract genes that exhibit signatures of positive selection and maintain species-specificity in sympatry, we will gain insight into the postmating prezygotic isolation genes that are ultimately driving speciation in this system.

In this study, we examine EPs in the *A. socius* complex to identify genes that show patterns of sequence evolution and lineage sorting that are consistent with a potential contribution to postmating prezygotic isolation between species. We examined seven EP coding genes, five of which were identified in a comparative proteome study between *A. socius* and *A. fasciatus* (*Marshall et al., 2011*), and two additional EP coding genes identified from unpublished EST libraries of *A. socius* accessory glands and testes (*Marshall et al., 2011*). The five EP coding genes from the comparative proteome study were previously tested for evidence of positive selection with limited population sampling and sequence fragments. When testing for evidence of selection in recently diverged species, biases can arise when a supposed fixation is in fact a polymorphism, when a true fixation is an ancestral polymorphism rather than a new mutation, and because neutral and adaptive mutations fix at different rates (*Keightley & Eyre-Walker, 2012*). We expanded population sampling across the species ranges for the two species that were tested previously, added the third species *A. sp. nov.* Tex, and examined longer sequence lengths to combat the biases that result from insufficient information. A major aim was to test whether our previous conclusions hold up to expanded sampling, at the population and species level and at the sequence level. Finally, we compared how these genes behave by looking at lineage sorting and allele sharing within and across the contact zone of *A. fasciatus* and *A. socius*. These combined analyses point toward an important role for some but not all examined EPs during the evolution of reproductive isolation within the *A. socius* complex of crickets.

## METHODS

### Background

Striped ground crickets of the *A. socius* complex inhabit moist grasslands across North America and do not show significant habitat isolation (*Howard, 1986*). The three species *A. socius*, *A. fasciatus*, and *A. sp. nov.* Tex form two contact zones, one between *A. fasciatus* (north) and *A. socius* (south) from Illinois to New Jersey (*Howard & Waring, 1991*), and one between *A. sp. nov.* Tex (west) and *A. socius* (east) near the Louisiana—Texas state line (*Traylor et al., 2008*). *A. fasciatus* and *A. socius* seem to have diverged from a common ancestor approximately 30,000 years ago, and *A. sp. nov.* Tex seems to have subsequently diverged from *A. socius* approximately 24,000 years ago (*Marshall, 2004*; *Marshall, 2007*). They have previously been shown to be isolated primarily via postmating prezygotic reproductive isolation (*Howard et al., 2002*; *Marshall, 2004*; *Marshall & DiRienzo, 2012*).

### Population and gene sampling

Crickets were collected from each population in the summer of 2010 and genotyped in the lab via allozymes (Isocytrate dehydrogenase and Hexokinase) to determine species identity (*Howard, 1983*; *Marshall, 2004*; *Traylor et al., 2008*). Sampling localities spanned the range of each species. The seven *A. socius* populations were sampled near Texarkana, AR (AR), Bottom, NC (Bot), Mt. Vernon, IL (IL), Pleasantville, NJ (Mi), Ruston, LA (LA), Gastonia, NC (NC), and Ardmore, OK (OK). The three *A. fasciatus* populations were sampled near Akron, OH (Akn), Frankfort, IL (FF), and New Paltz, NY (NP). The three *A. sp. nov.* Tex populations were sampled near Terrell, TX (Tx20), Royse City, TX (Tx30), and Gainesville, TX (Tx35). Contact zone populations of *A. fasciatus* and *A. socius* were sampled from two habitats at a single location in Kenna, WV. *A. fasciatus* was collected from a hillside habitat, which we call Kenna Hill (KH), and *A. socius* was collected along the base of hill near a creek which we call Kenna Creek (KC). We did not have samples from the contact zone between *A. socius* and *A. sp. nov.* Tex. General maintenance protocols followed *Marshall et al. (2009)*. Briefly, juveniles were reared to maturity in population and sex-specific plastic cages. All crickets were maintained at 27 °C and 14:10 h light to dark photoperiod.

We dissected male accessory glands and testes from three individuals per allopatric population and 9 individuals per contact zone population. cDNA was synthesized from each tissue using RNA isolated via an Ambion RNAqueous-4PCR (#AM1914) kit and standard protocols for 1st strand cDNA synthesis. General PCR and sequencing procedures followed *Marshall et al. (2011)*. Our reagent amounts for a 25 µL reactions were: 2.5 10× buffer B, 2.0 MgCl$_2$ (25 mM), 0.5 dNTP (10 mM), 0.5 for each primer (10 µM), 1 U Taq (Fisher), 0.5 cDNA, 18.3 ddH$_2$O. We used the following PCR program: 94 °C for 2:00 min, then 30 cycles of 94 °C for 30 s, annealing at 45–55 °C for 30 s, 72 °C for 1:00 min, and final elongation at 72 °C for 7:00 min. Specific annealing temperatures depended on individual primer melting temperatures (primers used are shown in Table S1). Sequencing was done at the Kansas State University Department of Plant Pathology DNA Sequencing and Genotyping Facility using Applied Biosystems BigDye$^{TM}$ chemistry on an Applied Biosystems 3730 DNA Analyzer.

We compared nucleotide sequences of seven EP coding genes: (1) *acg69* (ACG69), a novel protein of unknown function expressed in accessory glands; (2) *arginine kinase* (AK), a phosphagen kinase that catalyzes ATP-regeneration and energy transport in invertebrates and some protozoa (*Ellington, 2001*; *Noguchi, Sawada & Akazawa, 2001*; *Uda et al., 2006*); (3) *apolipoprotein A-1 binding protein* (APBP), a phosphoprotein expressed in sperm that is homologous to a mammalian sperm capacitation gene (*Jha et al., 2008*); (4) *ejaculate serine protease* (EJAC-SP), an abundant accessory gland-expressed serine protease previously shown to be involved in the induction of egg laying in successfully mated females (*Marshall et al., 2009*); (5) *serine protease inhibitor* (SPI), a testis-expressed serine-type endopeptidase inhibitor; (6) *aspartate aminotransferase* (GOT), a pyridoxal-phospate-dependent aminotransferase expressed in the testis that is also an allozyme historically used to diagnose species identity among *A. socius* complex crickets (*Howard, 1983*; *Howard, 1986*); (7) *sperm-associated antigen 6* (SPAG6), homologous to a mammalian protein important for sperm flagellar motility and the structural integrity of the central apparatus (*Neilson et al., 1999*; *Sapiro et al., 2002*). The first five genes were investigated to a lesser extent in a comparison of male ejaculate proteome profiles in *A. fasciatus* and *A. socius* (*Marshall et al., 2011*). On 2D-DIGE (differential in-gel electrophoresis) gels ACG69, EJAC-SP and SPI had non species-specific protein spots that indicate similar molecular weights, isoelectric points, and expression levels in the male ejaculate, while AK and APBP had species-specific spots that indicate differences in one or more of these proteomic traits. The latter two genes were EP coding genes present in EST libraries of *A. socius* accessory glands and testes (*Marshall et al., 2011*) that were identified as potential candidates based on a review of sperm biology literature.

Sequences formatted as haplotypes are available from NCBI GenBank PopSets 372477571 (ACG69), 372477483 (AK), 372477513 (APBP), 372477527 (EJAC-SP), 372477535 (GOT), 372477555 (SPAG6), 372477561 (SPI).

## Species tree-based analyses

We first applied tests of selection to the species tree of the *A. socius* complex (*A. fasciatus*, (*A. socius*, *A. sp. nov.* Tex)). The ratio of non-synonymous to synonymous substitution rates $\omega$ is widely used to detect signatures of selection acting upon protein coding genes (*Yang & Bielawski, 2000*; *Nielsen, 2001*; *Nielsen, 2005*; *Jensen, Wong & Aquadro, 2007*). When $\omega$ is larger than 1, positive or balancing selection is inferred. When $\omega$ is smaller than 1, negative or purifying selection is inferred. We aligned all sequences in BioEdit v.7.0.5.3 (*Hall, 1999*). We counted within species polymorphisms and between species fixations and calculated $\pi_s$, $\pi_a$, and $\theta = 4N_e\mu$ across all populations with DnaSP v.5.10.1 (*Librado & Rozas, 2009*). Unless stated otherwise, the majority of the remaining analyses were carried out in HyPhy v.2.2.1 (*Kosakovsky Pond, Frost & Muse, 2005*) or its online server Datamonkey (*Delport et al., 2010*). We selected a nucleotide substitution model (NucModelCompare.bf) at a model rejection level of 0.0002, the recommended level based on Bonferroni correction of comparing 203 increasingly parameter-rich nucleotide substitution models (*Kosakovsky Pond & Frost, 2005a*). Next we tested for evidence of recombination using the Genetic Algorithm Recombination Detection (GARD) method

(SingleBreakpointRecomb.bf). Recombination should be ruled out or accounted for because its presence can mislead inferences of selection due to no single phylogenetic tree being able to accurately describe the evolutionary relationship of recombinant sequences (*Anisimova, Nielsen & Yang, 2003*; *Shriner et al., 2003*; *Kosakovsky Pond et al., 2006*). GARD tests for evidence of recombination by comparing fits of a single tree vs. segment-specific trees to alignments. Segments are potential recombinant fragments, as defined by sequence in between possible breakpoints (variable sites).

Next we fit maximum likelihood models to the species tree to estimate $\omega$ at each branching node: the node between *A. fasciatus* and the other two species, and the node between *A. socius* and *A. sp. nov.* Tex. Using the codon substitution model MG94 (*Muse & Gaut, 1994*) while estimating codon frequencies based on combinations of nucleotide frequencies (called 3x4 in HyPhy), we fit and optimized a maximum likelihood model on the multiple sequence alignment for each gene, and jointly estimated both $\omega$ and their fine asymptotic normal CI estimates from the same likelihood model. HyPhy derives CI estimates analytically from the Fisher Information Matrices of the log likelihood surface of each model parameter (*Kosakovsky Pond & Muse 2005*).

Finally, we compared polymorphism within species to divergence between species at each branching node of the species tree. We compared $\omega$ for each gene using McDonald–Kreitman tests (*McDonald & Kreitman, 1991*) in DnaSP, and across all genes using a multilocus HKA test (*Hudson, Kreitman & Aguadé, 1987*) with the program HKA (*Wang & Hey, 1996*). The null model of the McDonald–Kreitman test is that the ratio of non-synonymous to synonymous intraspecific polymorphisms should be no different from the ratio of non-synonymous to synonymous fixations between species. The multilocus HKA test uses coalescent simulations to test the null model that all loci within a species will share the same effective population size, and loci between sister species will evolve under the same neutral mutation rate. The HKA program also reports an outlier observation that deviates the most from its expected value (*Wang & Hey, 1996*). The outlier corresponds to a specific locus and its level of polymorphism within species or fixation between species.

## Gene tree-based analyses

Gene trees show the evolutionary histories of individual genes and are less likely to concur with species trees the shorter the time since divergence due to incomplete lineage sorting and introgression (*Pamilo & Nei, 1988*; *Maddison, 1997*). Because genes more directly related to reproductive isolation and speciation should show patterns of evolution more closely resembling the species tree (*Wu, 2001*; *Feder, Egan & Nosil, 2012*), comparing gene trees should inform us of, or confirm, which genes are more likely to be the key genes involved in postmating prezygotic isolation, particularly when evidence suggests the action of selection upon internal nodes that separate incipient species.

Using Neighbor-Joining gene trees built using Tamura–Nei distances (*Tamura & Nei, 1993*) in HyPhy, we tested for evidence of selection on specific internal branches of the gene tree that had non-zero branch length (TestBranchdNdS.bf). This method estimates the non-synonymous rate while assuming a single synonymous substitution rate across a tree, so in effect tests for evidence of elevated $\omega$ on specific branches. We used the MG94 and

F81 (*Felsenstein, 1981*) substitution models and fit a single non-synonymous substitution rate across the tree with no within branch variation in $\kappa_s$ or $\kappa_a$. We compared this model to a model where the internal nodes of interest (those whose non-synonymous substitution rate's 95% CI did not overlap with zero based on a first pass run of this method on all internal branches with non-zero branch lengths) were allowed to vary from the global non-synonymous rate using likelihood ratio tests. We used MEGA6 (*Tamura et al., 2013*) to visualize these trees.

We tested for evidence of specific sites evolving under different selection regimes using the single likelihood ancestor counting (SLAC) method (*Kosakovsky Pond & Frost, 2005b*). SLAC (QuickSelectionDetection.bf) tests all sites of each gene for evidence of selection by first taking a tree and fitting a codon model (MG94 ×F81) to obtain a global estimate of $\omega$. It then reconstructs an ancestral sequence for each site across all internal nodes of the tree using joint maximum likelihood estimation, taking into account the estimate of $\omega$ obtained in the previous step. Then for each variable site, the method compares expected and observed numbers of synonymous and non-synonymous substitutions to detect selection.

Finally, we tested for evidence of episodic selection anywhere within each gene using a branch-site method. BUSTED (branch-site unrestricted statistical test for episodic diversification) tests whether there is evidence suggesting any single gene is under positive selection, while accounting for site-level variation in selection and variable selection on a subset of branches on a phylogenetic tree (*Murrell et al., 2015*). We partitioned branches into two categories: those of interest (foreground), which were the same branches from the branch method above (TestBranchdNdS) that had non-zero branch lengths, and the remainder (background). BUSTED optimizes the likelihood using a random effects likelihood framework (*Kosakovsky Pond et al., 2011*) and first fits an unconstrained model that proportions sites in the foreground branches into one of three variable rates including those over $\omega = 1$. This alternative model is then compared to a null model in which all rates are constrained to $\omega = {<}1$, but all sites are still fit and proportioned into three rates. These models are compared to each other using likelihood ratio tests.

### Evidence from the contact zone between *A. fasciatus* and *A. socius*

The genealogical sorting index (gsi) reflects the degree of lineage sorting of individual gene genealogies that occurs during speciation, with values ranging from zero (complete polyphyly) to 1 (complete monophyly) (*Cummings, Neel & Shaw, 2008*). We calculated gsi for each gene using the gsi web service (www.molecularevolution.org) with gene trees including both allopatric and contact zone individuals. We generated gene trees for gsi analysis with Neighbor Joining in TreeBeST v.1.9.2 (*Li, 2006*) using the option ntmm that calculates p-distances from codon alignments. We used this method because gsi requires a rooted tree and TreeBeST can use a species tree to root a gene tree using an algorithm (*Zmasek & Eddy, 2011*) that compares potential root positions on an unrooted tree and places the root where the differences between the gene tree and species tree are minimized (*Li, 2006*). The program permutes the labels (in our case, species identity) of the tips of the given tree multiple times (we used the default $n = 10,000$), each time determining the

gsi value of this new tree. The permuted *P*-value is the probability of randomly observing a gsi value equal to or better (higher) than the gsi value observed from the data. We used MEGA6 to visualize these trees.

Finally, we constructed statistical parsimony haplotype networks (*Templeton, Crandall & Sing, 1992*) of alleles from all three species to test for species-specificity of alleles. We used TCS v.1.21 (*Clement, Posada & Crandall, 2000*) to generate the haplotype networks using only allopatric individuals. Species-specific alleles were defined as those found only within each respective species. Common or shared alleles were those observed in more than one species. Once alleles were designated common or specific to a species, we looked at nine individuals each within the contact zone of *A. fasciatus* and *A. socius* and determined what types of allele these contact zone individuals possessed. As noted above, these individuals had previously been designated as fully (homozygous) *A. fasciatus* or *A. socius* based on allozymes. To compare the allelic distributions of these genes, we calculated a segregation metric, the dissimiliarity index *D* (*Duncan & Duncan, 1955*) using the R package SEG (*Hong, O'Sullivan & Sadahiro, 2014*). Duncan and Duncan's *D* is a measure of segregation in space that ranges from 0 (complete integration) to 1 (complete segregation). For genes for which data was missing (1 individual for the gene SPAG6), we calculated the potential range of *D* depending on the potential values for the missing observations.

## RESULTS

### Species tree-based analyses

We fit the nucleotide substitution model F81 to all genes, and found no evidence of recombination in any of our genes at any of the variable sites. We found relatively low levels of both synonymous and non-synonymous nucleotide variation within and among the *A. socius* complex species. The Watterson estimator $\theta$ ranged from 0.001 to 0.011, $\kappa_s$ ranged from 0.004 to 0.054, and $\kappa_a$ ranged from 0 to 0.009 (Tables 1 and 2). Next we estimated $\omega = \kappa_a/\kappa_s$ at each branching event of the species tree. In the older split between *A. fasciatus* and the other two species, the maximum likelihood estimates of $\omega$ exceeded 1, indicating evidence of positive or balancing selection in the genes AK ($\omega = 24.841$, 95 % CI [12.988–36.694]), APBP ($\omega = 13.495$, 95% CI [0–35.670]), and SPI ($\omega = 1.356$, 95 % CI [0–Inf]), while EJAC-SP approached $\omega = 1$ ($\omega = 0.993$, 95% CI [0–16.673]). In the younger split between *A. socius* and *A. sp. nov.* Tex, $\omega$ did not exceed 1 in any of the genes (Table 2). Because of the combination of low $\pi_s$, $\pi_a$, and $\theta$ and high $\omega$ for these genes, we will henceforth interpret $\omega > 1$ as evidence of positive selection as balancing selection is more likely to be accompanied by higher nucleotide diversity.

Not all genes had fixed non-synonymous substitutions between species and in these cases we were unable to apply the McDonald–Kreitman test (Table 2). For those genes that were testable, we did not find significant differences in $D_N/D_S$ compared to $P_N/P_S$ at either branching event (Fisher's exact test $P = 0.07$–1). We were unable to detect a significant departure from neutral expectations for the first branching event between *A. fasciatus* and the two other species using the multilocus HKA test ($X^2 P = 0.916$). We did detect a significant departure from neutral expectations for the second branching

**Table 1  Nucleotide variation within each *A. socius* complex species.**

| Gene | Length | *A. fasciatus* | | | | *A. socius* | | | | *A. sp. nov.* Tex | | | |
|---|---|---|---|---|---|---|---|---|---|---|---|---|---|
| | | *n* (pop) | $\pi_s$ | $\pi_a$ | $\theta_{fas}$ | *n* (pop) | $\pi_s$ | $\pi_a$ | $\theta_{soc}$ | *n* (pop) | $\pi_s$ | $\pi_a$ | $\theta_{Tex}$ |
| ACG69 | 414 | 9 (3) | 0.005 | 0.004 | 0.007 | 14 (6) | 0.021 | 0.009 | 0.011 | 7 (3) | 0 | 0 | 0 |
| AK | 1,173 | 9 (3) | 0.002 | <0.001 | 0.001 | 15 (6) | 0.004 | 0.001 | 0.002 | 6 (3) | 0.003 | 0.001 | 0.002 |
| APBP | 705 | 9 (3) | 0.001 | 0 | 0.001 | 15 (5) | 0.005 | 0 | 0.001 | 8 (3) | 0 | 0 | 0 |
| EJAC-SP | 726 | 9 (3) | 0 | 0 | 0 | 16 (6) | 0.001 | <0.001 | 0.001 | 9 (3) | 0.003 | 0 | 0.001 |
| GOT | 1,122 | 9 (3) | 0.002 | 0 | <0.001 | 17 (7) | 0 | 0 | 0 | 9 (3) | 0.005 | 0.001 | 0.002 |
| SPAG6 | 426 | 9 (3) | 0 | 0 | 0 | 17 (6) | 0 | 0 | 0 | 8 (3) | 0.005 | 0 | 0.001 |
| SPI | 315 | 9 (3) | 0.007 | 0 | 0.002 | 16 (6) | 0 | 0 | 0 | 9 (3) | 0.008 | 0.001 | 0.002 |

**Notes.**

The number of individuals sampled for each gene are shown with the number of populations they came from (*n* (pop)).

$\pi_s$, the average number of synonymous nucleotide differences per site for any random pair of sequences; $\pi_a$, the average number of non-synonymous nucleotide differences per site for any random pair of sequences; $\theta$, a metric of the population substitution rate; all metrics were calculated across populations.

event between *A. socius* and *A. sp. nov.* Tex. ($X^2 P = 0.012$). The outlier that diverged most from null expectations was polymorphism in GOT in *A. sp. nov.* Tex., but coalescent simulations were unable to determine that this was a significant difference ($P = 0.06$). Both the McDonald–Kreitman and HKA tests are likely to be underpowered for our species given the age and amount of variation present in the *Drosophila* systems that both tests were originally developed for *Hudson, Kreitman & Aguadé (1987 )*; *McDonald & Kreitman (1991)*; *Wang & Hey, (1996)*.

## Gene tree-based analyses

We tested for evidence that any of the internal branches of each gene tree were evolving at a higher non-synonymous rate than the other branches (Fig. 1). We found evidence that the model in which two internal nodes in the gene tree of AK were allowed to evolve at a variable non-synonymous rate was a better fit than the single rate class model ($P = 0.004$). These nodes evolved at non-synonymous substitution rates 26.331 (95% CI [4.377–81.378]) and 0.147 (95% CI [0.008–0.649]), while the shared non-synonymous substitution rate was estimated to be 0. The first node (5) is the node that separates *A. fasciatus* and the two other species on the AK gene tree. The second node (43) is the node that separates *A. socius* and *A. sp. nov.* Tex on the AK gene tree. We also found evidence that two internal nodes in the gene tree of APBP were evolving at a different rate compared to the background branches ($P = 0.015$). These nodes were evolving at rates 43.793 (95% CI [2.515–194.273]) and 19.008 (95% CI [1.088–84.035]), while the shared non-synonymous substitution rate was estimated to be 0. The first node (11) is the node between *A. fasciatus* and the two other species on the APBP gene tree. The second node (45) is the node between *A. socius* and *A. sp. nov.* Tex on the APBP gene tree. Despite evidence that some internal nodes were evolving at a higher rate compared to background nodes in the other genes, we did not have sufficient evidence to suggest branches in any of the other gene trees were evolving at more than one non-synonymous rate class (ACG69 (6 tested) $P = 0.171$, EJAC-SP (1 tested) $P = 0.277$, GOT (3 tested) $P = 0.218$, SPI (1 tested) $P = 0.265$, SPAG6 (2 tested) $P = 1$).

Noh and Marshall (2016), *PeerJ*, DOI 10.7717/peerj.1678

**Table 2** Nucelotide variation at each branching node of the *A. socius* complex species tree.

| Gene | Length | Between *A. fasciatus* & (*A. socius* + *A. sp. nov.* Tex) | | | | | | | Between *A. socius* & *A. sp. nov.* Tex | | | | | | |
|---|---|---|---|---|---|---|---|---|---|---|---|---|---|---|---|
| | | $P_N$ | $P_S$ | $D_N$ | $D_S$ | $\kappa_s$ | $\kappa_a$ | $\omega$ (95% CI) | $P_N$ | $P_S$ | $D_N$ | $D_S$ | $\kappa_s$ | $\kappa_a$ | $\omega$ (95 % CI) |
| ACG69 | 414 | 7 | 6 | 0 | 0 | 0.024 | 0.009 | 0.572 (0, 1.366) | 7 | 6 | 0 | 0 | 0.021 | 0.007 | 0.142 (0, 0.420) |
| AK | 1,173 | 3 | 12 | 2 | 0 | 0.006 | 0.003 | 24.841 (12.988, 36.694) | 2 | 9 | 1 | 2 | 0.011 | 0.002 | 0.147 (0, 0.436) |
| APBP | 705 | 1 | 4 | 1 | 0 | 0.005 | 0.003 | 13.495 (0, 35.670) | 0 | 3 | 1 | 0 | 0.007 | 0.002 | 0.285 (0, 0.843) |
| EJAC-SP | 726 | 1 | 3 | 0 | 0 | 0.008 | 0.001 | 0.993 (0, 16.673) | 1 | 2 | 0 | 1 | 0.014 | 0.002 | 0.158 (0, 0.467) |
| GOT | 1,122 | 3 | 7 | 1 | 1 | 0.011 | 0.002 | 0.274 (0, 0.811) | 3 | 4 | 0 | 2 | 0.011 | 0.001 | 0.136 (0, 0.402) |
| SPAG6 | 426 | 0 | 1 | 0 | 2 | 0.029 | 0 | 0 (0, 0.001) | 0 | 1 | 0 | 0 | 0.004 | 0 | 0.001 (0.001, 0.002) |
| SPI | 315 | 3 | 4 | 0 | 0 | 0.02 | 0.003 | 1.356 (0, inf) | 1 | 1 | 2 | 3 | 0.054 | 0.009 | 0.178 (0, 0.425) |

**Notes.**

$P_N$, Non-synonymous polymorphisms; $P_S$, synonymous polymorphisms; $D_N$, Non-synonymous fixations; $D_S$, synonymous fixations; $\kappa_s$, Rate of non-synonymous substitutions per non-synonymous site; $\kappa_a$, Rate of synonymous substitutions per synonymous site; $\omega = \kappa_a/\kappa_s$, Estimate from maximum likelihood model fit in HyPhy, along with 95% asymptotic normal confidence intervals (CI).

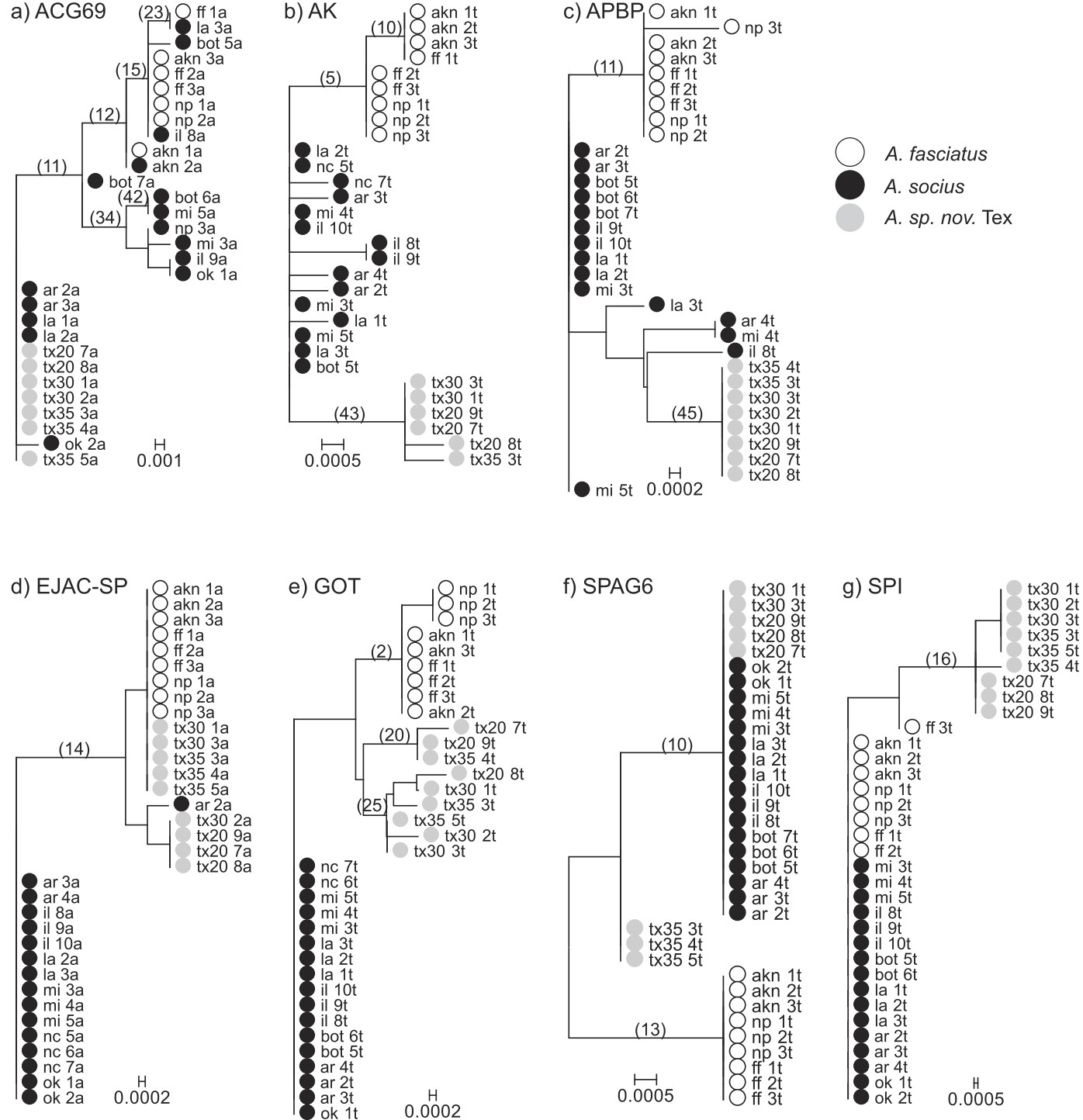

**Figure 1** **Neighbor-Joining gene trees that include allopatric individuals only.** Internal branches whose non-synonymous substitution rate confidence intervals did not overlap with zero are marked. These internal branches were tested for evidence of elevated rates of non-synonymous substitutions (TestBranchdNdS) and for episodic selection among sites and branches (BUSTED).

**Table 3  Genealogical sorting index values based on individual rooted gene trees.** Values range from zero (complete polyphyly) to one (complete monophyly). $P$-values are obtained from 10,000 permutations that randomly reassign tip labels on the tree and represent the probability of observing a tree with a gsi value more extreme than the current toplogy.

| Gene | gsi-fas | $P_{perm}$ | gsi-soc | $P_{perm}$ | gsi-Tex | $P_{perm}$ |
|---|---|---|---|---|---|---|
| ACG69 | 0.458 | <0.001 | 0 | 1 | 0.836 | <0.001 |
| AK | 1 | <0.001 | 0.843 | <0.001 | 1 | <0.001 |
| APBP | 1 | <0.001 | 0.849 | <0.001 | 1 | <0.001 |
| EJAC-SP | 0.609 | <0.001 | 0.730 | <0.001 | 0.288 | 0.001 |
| GOT | 0.844 | <0.001 | 0.533 | <0.001 | 1 | <0.001 |
| SPAG6 | 1 | <0.001 | 0.667 | <0.001 | 0.191 | 0.016 |
| SPI | 0.189 | 0.010 | 0.622 | <0.001 | 1 | <0.001 |

Using SLAC and comparing model fits of nucleotide substitution models and codon models, for most genes (ACG69, AK, APBP, SPI, SPAG6) we found no evidence of any specific sites evolving under positive or negative selection. For EJAC-SP, one site (position 115) in which the amino acid serine was maintained through synonymous substitutions, showed evidence of negative selection ($P = 0.011$). Also for GOT, one site (position 88) in which the amino acid threonine was maintained through synonymous substitutions, showed evidence of negative selection ($P = 0.037$).

We used BUSTED to test for episodic selection anywhere in each gene. We tested whether any sites evolved at a faster $\omega$ rate within the internal branches with non-zero branches tested in the previous branch analysis. We found non-significant evidence of episodic selection only in ACG69 using this method (Likelihood Ratio Test $P = 0.062$). The unconstrained model suggests 1.7% of foreground branch sites may be evolving with $\omega = 318.77$ while background branch sites have a maximum $\omega = 0.96$ (100% of the sites were partitioned into this rate class). This branch-site test is likely to be underpowered for our species given that the numbers of parameters that were estimated for the unconstrained model was upwards of 80 (depending on the number of terminal taxa for each gene), while the difference in numbers of parameters between the complex model and null model was only 1. We do not discuss results from this method further because results were non-significant across all genes.

### Evidence from the contact zone between *A. fasciatus* and *A. socius*

Comparisons of genealogical sorting index values based on gene trees including all sampled individuals, both allopatric and contact zone, indicated that lineage sorting was ongoing as gsi values ranged from 0 to 1, with a median value of 0.836. If we use the median value as a cutoff, only AK and APBP showed relatively advanced lineage sorting for all three species while GOT showed advanced lineage sorting for *A. fasciatus* and *A. sp. nov.* Tex but not *A. socius* (Table 3 and Fig. 2).

The statistical parsimony haplotype networks generated using allopatric individuals of all three species showed that alleles in AK, APBP, and GOT were specific to each species, while the alleles present in other genes included both species-specific alleles and alleles

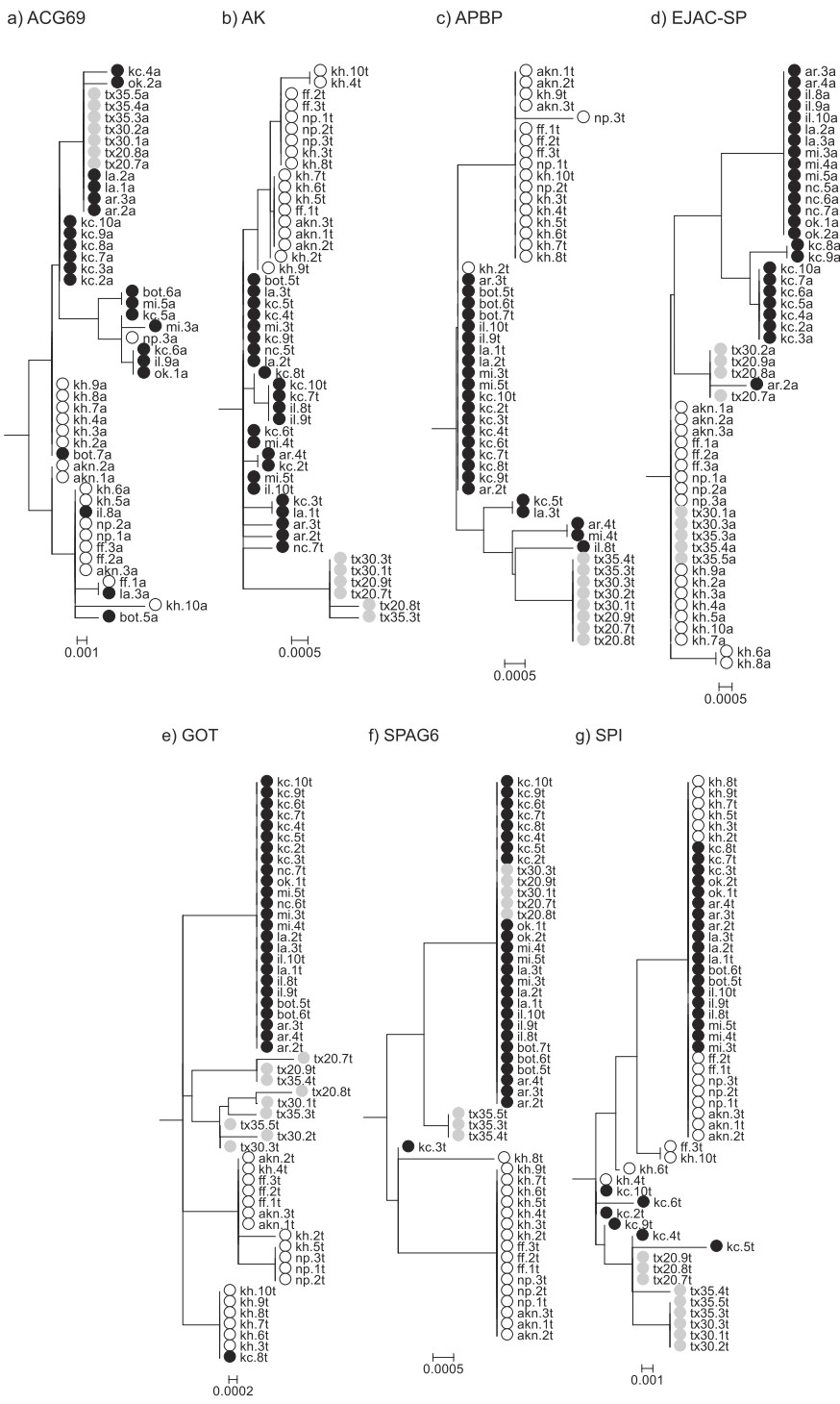

**Figure 2   Rooted Neighbor-Joining gene trees that include both allopatric and contact zone individuals.** These trees were used to estimate degrees of lineage sorting for each species.

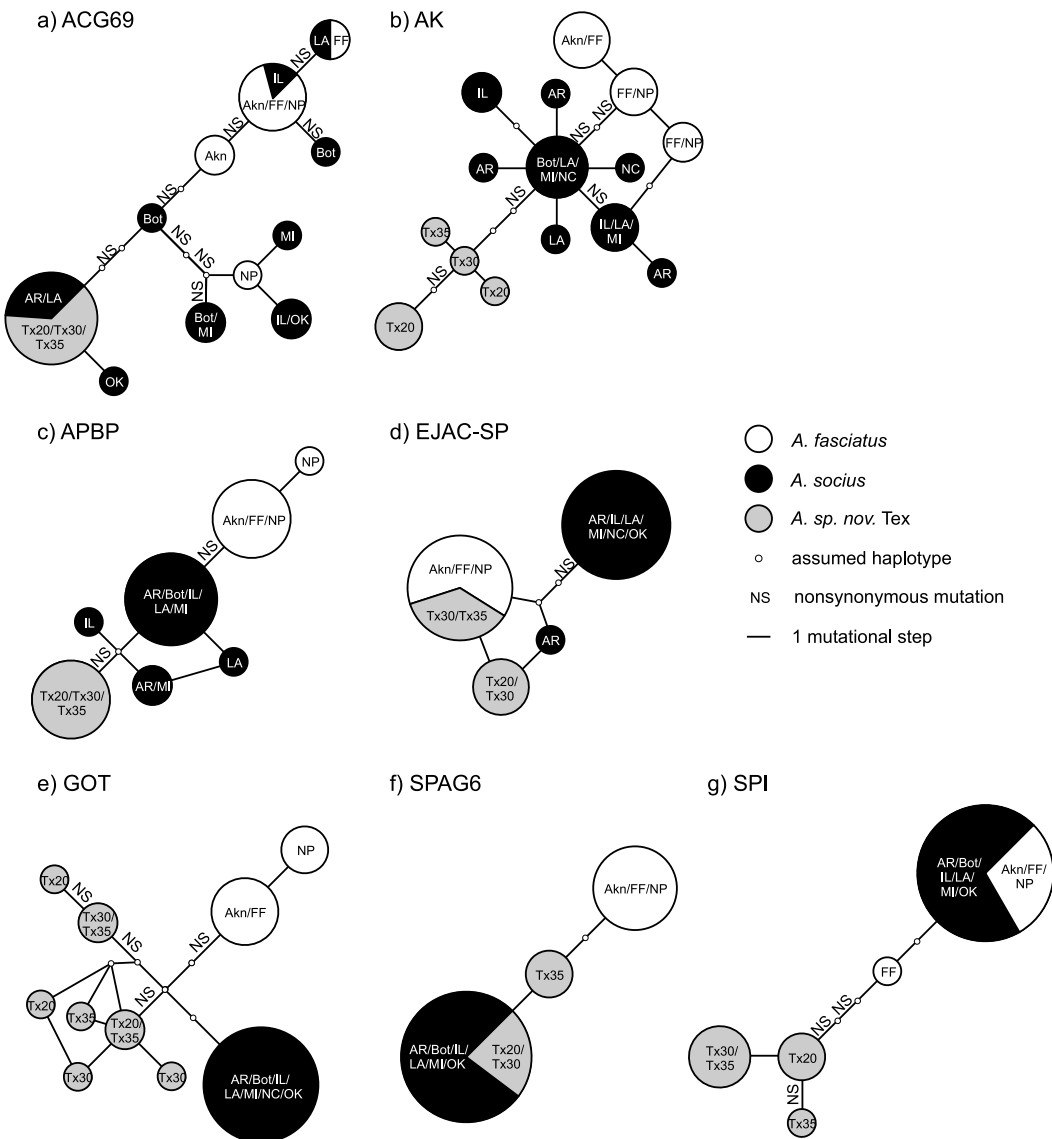

**Figure 3** **Statistical parsimony haplotype networks including only allopatric individuals, with mutational steps and nonsynonymous substitutions indicated.** Population abbreviations are as in the main text. Sizes of each haplotype correspond to the relative number of individuals that possessed each haplotype.

shared between two species (Fig. 3). Within the contact zone between *A. fasciatus* and *A. socius*, all genes except APBP had a mix of alleles that we had already observed in allopatric individuals as well as new alleles that were specific to the contact zone (Fig. 2). We designated these new alleles as species-specific based on their allelic distributions among contact zone individuals and calculated the degree of allelic segregation within the contact zone. The dissimilarity index $D$ values we observed were bimodally distributed and indicated allelic segregation was higher in the contact zone for AK, APBP, EJACSP and SPAG6 ($D = 0.771$–$1$) and lower in ACG69, GOT and SPI ($D = 0.164$–$0.353$) (Fig. 4).

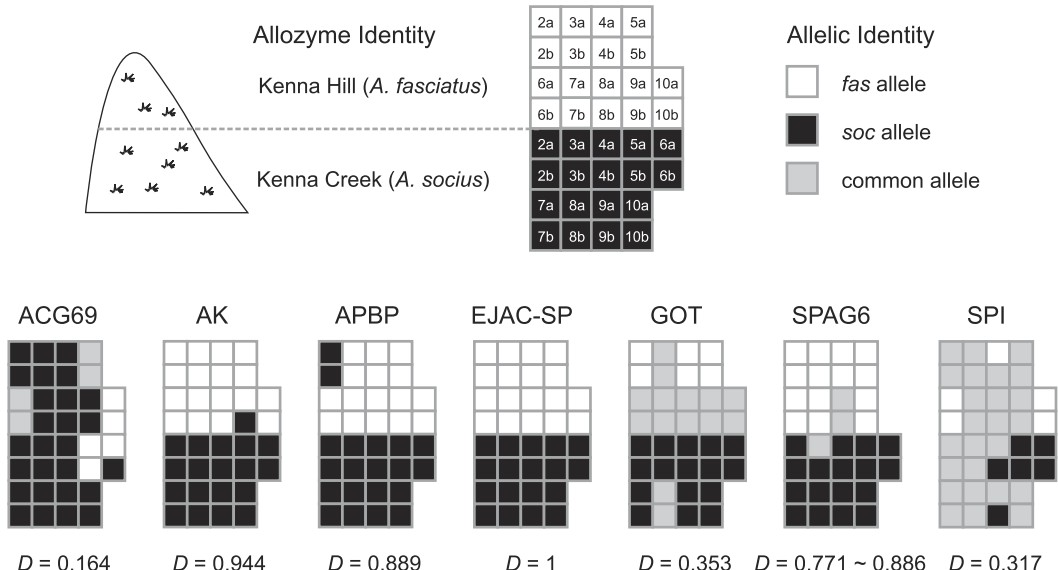

**Figure 4** **Distribution of species-specific vs. common (shared) alleles within the *A. fasciatus* and *A. socius* contact zone in Kenna, WV.** Nine individuals each with allozyme identities of pure (homozygous) *A. fasciatus* and *A. socius* had varying patterns of allelic identities for the seven genes. Numbers (2–9) indicate the sampled individual and letters (A and B) indicate the alleles within each individual. The dissimilarity index *D* is a metric of the degree of allelic segregation between the two contact zone populations.

## DISCUSSION

Studying speciation in recently diverged species is attractive because we can potentially identify variation that is associated with divergence and not accumulated after (*Andrés et al., 2013*). But using within and between species patterns of sequence variation to detect adaptive evolution is particularly challenging in recently diverged species because less evolutionary time has passed to allow for fixed differences to accumulate between incipient species relative to within species polymorphism (*Keightley & Eyre-Walker, 2012*). We gathered data from populations across each species distribution so as to accurately differentiate between polymorphisms and fixations and detect new mutations to confirm and expand upon previous results from more limited population sampling (*Marshall et al., 2011*). Combining evidence from multiple tests allowed us to identify the most likely putative postmating prezygotic isolation genes among our EP coding genes (Table 4). Ultimately, the methodological approaches that we were able to apply most successfully were those that took advantage of evolutionary processes specific to young species: lineage sorting in gene trees and allele sharing in haplotype networks.

Data from the relatively recently (∼30,000 years) diverged *A. socius* species complex showed a general lack of both synonymous and non-synonymous nucleotide variation within and among all investigated genes (Tables 1 and 2). Our estimates of sequence variation were also at least an order of magnitude smaller compared to other known estimates from accessory gland protein coding genes in various other species groups (*Mueller et al., 2005*; *Wagstaff & Begun, 2005*; *Almeida & DeSalle, 2008*), including some *Gryllus* crickets whose species are of roughly similar age (*Andrés et al., 2006*). Since

**Table 4  Summary of results found testing for evidence of selection in ejaculate protein coding genes in the *A. socius* complex.**  The specific method used for each result is shown in parentheses.

| Genes | Species tree $\pi_s, \pi_a, \theta$ | Species tree $\omega$ (MLE) | $D_N/D_S > P_N/P_S$ (MK) | Departure from other genes (HKA) | Gene tree evidence of branch selection (Test-BranchdNdS) | Gene tree evidence of site selection (SLAC) | Lineage sorting (allopatric + contact zone) (gsi) | Allelic segregation in contact zone (*D*) |
|---|---|---|---|---|---|---|---|---|
| ACG69 | Small | All <1 | NA | No | No | No | Low | Low |
| AK | Smaller | Some >1 | No | No | Yes | No | High | High |
| APBP | Smaller | Some >1 | No | No | Yes | No | High | High |
| EJAC-SP | Smaller | Some ~1 | NA | No | No | Purifying selection | Low | High |
| GOT | Smaller | All <1 | No | Maybe | No | Purifying selection | Moderate | Low |
| SPAG6 | Smaller | All <1 | NA | No | No | No | Low | High |
| SPI | Smaller | Some >1 | No | No | No | No | Low | Low |

speciation in the *A. socius* complex is thought to coincide with glaciation history (*Marshall, 2004*; *Marshall, 2007*) population bottlenecks may have contributed to sequence variation patterns. Our results would have benefited from increased within-population sampling so that metrics such as Tajima's or Fu and Li's *D* and site frequency spectra, as well as estimates of population size and recombination rates (*Nielsen, 2005*) could be accurately applied. Unsurprisingly, the tests we used that relied on sequence variation on the known species tree had minor success identifying genes with evidence of positive selection based on $\omega$ (Table 4). The maximum likelihood estimates of $\omega$ on the species tree indicated that AK, APBP, EJAC-SP and SPI showed evidence of positive selection at the branching event between *A. fasciatus* and the other two species (Table 2). However, McDonald–Kreitman tests were not applicable to several genes and HKA tests were generally inconclusive.

We found approaches that used gene trees and haplotype networks were more successful at detecting evidence of positive selection, and these approaches were particularly powerful within the contact zone (Table 4). When allopatric gene trees were tested for evidence of internal branches evolving at different rates compared to the rest of the tree, we found evidence that AK and APBP both had internal branches that were evolving at higher non-synonymous substitution rates (Fig. 1). These internal branches respectively separated *A. fasciatus* from the other two species, and *A. socius* from *A. sp. nov.* Tex. When all allopatric and contact zone individuals were examined, the genealogies of AK and APBP indicated that these genes were relatively advanced in their degrees of lineage sorting in all three species of the *A. socius* complex compared to the other genes (Table 3 and Fig. 2). GOT also showed advanced lineage sorting, but only in the two species *A. fasciatus* and *A. sp. nov.* Tex. Within the contact zone of *A. fasciatus* and *A. socius*, AK, APBP, EJAC-SP, and SPAG6 showed highly segregated allelic distributions (Fig. 4). The two genes with intermediate lineage sorting or allelic segregation patterns, EJAC-SP and GOT, showed evidence of purifying selection acting upon specific sites, as might be expected of most functional protein coding genes (*Lawrie et al., 2013*), but no evidence of positive selection.

The overlap among these results suggests AK and APBP are the most likely candidates for postmating prezygotic isolation genes (Table 4). While the remaining genes that show intermediate patterns may still contribute to reproductive isolation, they are less likely to be the major contributors to isolation barriers that act in a species-specific manner.

These patterns fit models of ongoing speciation in the face of gene flow, where incomplete lineage sorting and introgression are major confounding factors (*Machado & Hey, 2003*; *Broughton & Harrison, 2003*; *Payseur, 2010*). Speciation genes are more likely to become fixed for species-specific alleles early in the process of speciation, and therefore are expected to be relatively exempt from incomplete sorting and subject to reduced introgression (*Wu, 2001*; *Feder, Egan & Nosil, 2012*). Similar patterns have been observed in *Drosophila*, field crickets, and moths (*Ting, Tsaur & Wu, 2000*; *Dopman et al., 2005*; *Maroja, Andrés & Harrison, 2009*; *Andrés et al., 2013*; *Larson et al., 2013*). It is possible that these genes are not the direct targets but rather linked to targets of divergent selection as the interaction between linkage and selection makes it challenging to distinguish between recurrent positive selection, background selection, and Hill–Robertson effects (*Hill & Robertson, 1966*; *Charlesworth, 1994*; *Andolfatto, 2007*; *Charlesworth et al., 2009*). Because both genes were identified as candidates through comparative proteomics (*Marshall et al., 2011*) this seems relatively unlikely, but the genomic regions around these genes should be investigated for evidence of selective sweeps to rule out this possibility.

Many studies of reproductive proteins report evidence of positive selection acting on a subset of the genes examined, in both males (*Begun et al., 2000*; *Swanson et al., 2001*; *Clark & Swanson, 2005*; *Wagstaff & Begun, 2005*; *Andrés et al., 2006*; *Karn et al., 2008*; *Ramm et al., 2008*; *Almeida & DeSalle, 2008*; *Walters & Harrison, 2010*) and females (*Swanson et al., 2004*; *Panhuis & Swanson, 2006*; *Lawniczak & Begun, 2007*; *Prokupek et al., 2008*; *Kelleher & Markow, 2009*; *Kelleher, Clark & Markow, 2011*). However, there are few examples of adaptive reproductive protein evolution leading to reproductive isolation outside of gamete recognition proteins (e.g., *Geyer & Palumbi, 2003*; *McCartney & Lessios, 2004*; *Springer & Crespi, 2007*). Our putative postmating prezygotic isolation genes AK and APBP are two of the first examples of sperm maturation and capacitation related proteins that show evidence of fixed non-synonymous substitutions between species-specific alleles leading to reproductive isolation (Fig. 3). We had previously observed this pattern between *A. fasciatus* and *A. socius* for both AK and APBP (*Marshall et al., 2011*), but finding the same pattern in the mutational steps between *A. socius* and *A. sp. nov.* Tex with different species-specific non-synonymous substitutions, with expanded population and sequence sampling, emphasizes the potential importance of these candidates.

Whether there are functional consequences to the species-specific non-synonymous substitutions in AK and APBP needs to be investigated further. Both candidates are homologous to mammalian sperm capacitation proteins. Sperm maturation and capacitation occur inside the female reproductive tract of mammals (*Visconti et al., 2011*). In insects and nematodes, sperm are capacitated and become motile by serine proteases present in the seminal fluid (*LaFlamme & Wolfner, 2013*). This process occurs within the spermatophores of Lepidoptera (*Osanai & Chen, 1993*; *Friedländer, Jeshtadi & Reynolds, 2001*), and in the seminal vesicles of *Drosophila* who do not make spermatophores (*Osanai*

*& Chen, 1993*). In either case, most insect sperm should already be mature and capacitated once spermatophores are transferred to females. Therefore the specific hypothetical function of these genes should be related to a different postcopulatory process that occurs within the female reproductive tract.

Female insects store sperm in specialized organs such as spermathecae and seminal receptacles after copulation, often for prolonged periods of time (*Schnakenberg, Siegal & Bloch Qazi, 2012*). Sperm lose motility within a day in flies, unless stored by females into these organs (*Schnakenberg, Matias & Siegal, 2011*). Conspecific sperm precedence among *Drosophila* species has been shown to involve post-copulatory processes that occur within the female reproductive tract that include sperm transfer, the displacement and ejection of less preferred (heterospecific) sperm, and the selective use of preferred (conspecific) sperm from different storage organs for fertilization (*Manier et al., 2013a*; *Manier et al., 2013b*). In flies, spermathecal secretory cells are intimately involved in the female driven part of these processes (*Schnakenberg, Matias & Siegal, 2011*). Endopeptidases produced by these secretory cells are necessary for recruiting sperm to spermathecae and also maintaining sperm motility in the seminal receptacle. The same endopeptidases also affect egg laying, so that females laid fewer eggs when their secretory cells were ablated. More recently, it was shown that the number of secretory cells that produce these endopeptidases determined whether female flies ovulated and layed eggs (*Sun & Spradling, 2013*). It is yet unknown whether sequence variation in these endopeptidases is related to variation in female fecundity. Other potential roles for the secretions of these cells as related to sperm precedence include a chemotactic function that would attract select sperm to different locations of the female reproductive tract (*Wolfner, 2011*).

The specific mechanism of conspecific sperm precedence that APBP is involved in may be related to the appropriate phosphorylation state of the phosphoprotein (APBP) depending on the female vs. male species combination. APBP becomes phosphorylated during murine sperm capacitation and co-localizes with cholesterol during this process, but its specific function is unknown (*Jha et al., 2008*). As noted above, insect sperm do not undergo capacitation as mammalian sperm do. As a putative binding protein with an enzymatic function, hypothetical functions of APBP in the female reproductive tract include that as a signal for sperm to be transferred to the preferred storage organ. It may also be involved in a signaling cascade that induces ovulation and oviposition in females. Between *A. fasciatus* and *A. socius*, when females were mated with only a single male those mated with heterospecific males laid fewer eggs but an equal proportion of fertilized eggs compared to females mated with conspecific males (*Gregory & Howard, 1993*).

Insects and other ecdysozoans possess AK as their sole phosphagen system for cellular energy metabolism, and accordingly, arginine phosphate and its phosphagen kinase AK are found primarily in muscles, but also in sperm and compound eyes (*Strong & Ellington, 1993*; *Kucharski & Maleszka, 1998*; *Ellington, 2001*). The possible roles of AK as an EP can be related to sperm motility or the acrosome reaction (*Strong & Ellington, 1993*; *Niksirat et al., 2015*). As an energy-related phosphagen kinase, AK may confer to sperm an enhanced ability to move toward a sperm storage organ and resist displacement and ejection by females. In honeybees, AK is expressed in both male seminal fluid (*Baer et al.,*

*2009b*) and female spermathecal fluid (*Baer et al., 2009a*). When tested over a two-year period AK enzymatic activity decreased in mated female spermathecal fluid and sperm motility also decreased (*Al-Lawati, Kamp & Bienefeld, 2009*). The proteomes of freshly ejaculated vs. stored honeybee sperm are also known to differ, particularly in terms of reduced activity of glycolytic enzymes that should be used for energy metabolism (*Poland et al., 2011*). Poland and colleagues note that most of the same enzymes are also present in the spermathecal fluid (*Baer et al., 2009a*), suggesting female physiology has an active role in maintaining stored sperm. Two structural loops and several active sites near them are the proposed interaction interface of AK with the guanidinium groups of its substrates (*Zhou et al., 1998*; *Pruett et al., 2003*; *Azzi et al., 2004*; *Clark, Davulcu & Chapman, 2012*). As might be expected for an integral enzyme, the non-synonymous substitutions we observed do not occur at these specific sites, though they may still influence its activity.

## CONCLUSIONS

*A. socius* complex crickets provide an excellent opportunity to identify patterns of evolution in speciation genes for two major reasons: speciation is incomplete as evidenced by ongoing gene flow in the field, and isolation is through a single type of reproductive isolation barrier, i.e., postmating prezygotic phenotypes. We found that when speciation is ongoing, combining multiple approaches, particularly those that incorporate evidence from gene trees and haplotype networks, was important for identifying putative postmating prezygotic isolation genes. Both AK and APBP have fixed, or nearly fixed, non-synonymous substitutions between both *A. fasciatus* and the other two species, and between *A. socius* and *A. sp. nov.* Tex. Both genes showed advanced lineage sorting across and within the contact zone, and allelic segregation within the contact zone. The next step is to determine the functional consequence of allelic variation in either EP in conspecific sperm precedence and the successful induction of female oviposition in the *A. socius* complex.

## ACKNOWLEDGEMENTS

We thank Christopher Garcia for collecting the crickets used in this study and Diana Huestis and Shanda Wheeler for their help in isolating RNA and screening individuals for species status with allozymes. We also thank Daniel J. Howard for his support throughout this work. This is contribution no. 12-015-J from the Kansas Agricultural Experiment Station.

### Funding

Funding from the National Science Foundation to JLM (DEB-0746316) supported this work. The funders had no role in study design, data collection and analysis, decision to publish, or preparation of the manuscript.

## Grant Disclosures

The following grant information was disclosed by the authors:
National Science Foundation: DEB-0746316.

## Competing Interests

The authors declare there are no competing interests.

## Author Contributions

- Suegene Noh performed the experiments, analyzed the data, contributed reagents/materials/analysis tools, wrote the paper, prepared figures and/or tables, reviewed drafts of the paper.
- Jeremy L. Marshall conceived and designed the experiments, analyzed the data, contributed reagents/materials/analysis tools, wrote the paper, reviewed drafts of the paper.

## DNA Deposition

The following information was supplied regarding the deposition of DNA sequences:
  Sequences formatted as haplotypes are available from NCBI GenBank PopSets 372477571 (ACG69), 372477483 (AK), 372477513 (APBP), 372477527 (EJAC-SP), 372477535 (GOT), 372477555 (SPAG6), 372477561 (SPI).

## Data Availability

  Two archives are included in the submission. The content files can be extracted using the following command in a Unix shell:
  tar-xvzf archive_file_name.

## Supplemental Information

Supplemental information for this article can be found online at http://dx.doi.org/10.7717/peerj.1678#supplemental-information.

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
