# Peer review of "Sorted gene genealogies and species-specific nonsynonymous substitutions point to putative postmating prezygotic isolation genes in Allonemobius crickets"

_PeerJ, doi:10.7717/peerj.1678_

## Round 0.1 · original submission · Major Revisions

It appears that this ms is largely a repetition of your 2011 study, reaching similar conclusions. Please highlight more clearly why repeating the study is warranted and what additional insights are gained.

The ms also requires some clarification concerning the methods - as clearly outlined by the reviewers.

Reviewer 1 ·

Basic reporting

No comments

Experimental design

No comments

Validity of the findings

No comments

Additional comments

The manuscript, “Sorted gene genealogies and species-specific nonsynonymous substitutions point to putative postmating prezygotic isolation genes” is a detailed study of the patterns of divergence for several ejaculate proteins. The analyses are appropriate and the authors present a strong case that two of these genes are candidate barrier genes. I reviewed a previous version of this manuscript and the authors addressed all of my earlier comments.

Reviewer 2 ·

Basic reporting

No comments

Experimental design

Lines 131-132: Authors should present evidence that these two markers to reliably identify all A. fasciatus and A. socius in both sympatry and allopatry. A. sp. nov. Tex was not in Howard 1983 and 1986 (unless it has been renamed), so evidence for these markers to differentiate A. sp. nov. Tex from A. fasciatus and A. socius is needed.

Line 133: When were samples collected?

Line 142: Please include maintenance procedures.

Lines 143-144: ACGs and testes were dissected from three individuals per allopatric
population and 9 individuals per contact zone population. Phasing was done for all genes but APBP (Lines 219-220). After phasing, there should therefore be 6 alleles per allopatric population and 18 alleles per contact zone population for all genes but APBP. There were 7 allopatric A. socius populations, 3 allopatric A. fasciatus populations, 3 allopatric A. sp. nov. Texas populations, 1 contact zone A. fasciatus population and 1 contact zone A. socius population.

If I have interpreted your sampling design correctly, I do not understand how the data are presented in Table 1. The authors are well aware that nucleotide variation and Tajima's D and Fu and Li's D are population-level statistics, but I don't see populations denoted in Table 1. What does "n" refer to - number of alleles or number of individuals? If populations were grouped together for these analyses, please explain which populations are represented in Table 1. Please also justify this decision.

Tajima's D and Fu and Li's D, if applied to each population separately, would have very limited statistical power in allopatric populations (6 alleles per gene per population, and only 3 alleles per population for APBP). Please justify use of these statistics on such small data sets, if populations were analyzed separately.

Line 146-147: Please include PCR and sequencing procedures.

Line 147: Please include amounts and concentrations of PCR reagents.

Line 150-151: Please explain what is meant by species-specific proteome profiles.

Line 161-163: Please give more information on why SPAG-6 was chosen. Why is "sperm flagellar motility and the structural integrity of the central apparatus" a candidate function for involvement in reproductive isolation? There are many proteins involved in sperm flagellar motility - why was SPAG-6 chosen in particular, over other proteins with similar function?

Line 164: Please explain what is meant by non species-specific proteome profiles.

Lines 176-178: Data from two control genes are not sufficient to separate demographic signatures from selection signatures on the site frequency spectrum when using Tajima's D and Fu and Li's D. The two control genes (SPI, ACG69) have strongly negative D values in A. fasciatus, and lack of statistical significance may be related to small sample size. Strongly negative D values could indicate demographic non-equilibrium in A. fasciatus, which would mean that D p-values based on beta distributions in DnaSP are inappropriate. Strongly negative D values in A. fasciatus could also indicate selection acting on SPI and ACG69, in which case these are not appropriate control genes. However, I don't know how the data are presented in Table 1 (see my comments on Lines 143-144), so my interpretation of Table 1 might change if I had this information. The authors should clarify Table 1 and justify the use of two control genes to represent genome-wide demographic effects.

Also, control genes were chosen because the proteins they encode had non species-specific proteome profiles, not because they were presumed to be free from the influences of natural selection. The authors should explain their rationale for choice of control genes in the context of suitability for population genetic analyses.

Lines 178-180: The usage of the HKA statistic should be clarified. In Lines 235-236, the authors write, "…we used a standard multilocus HKA test and HKA outlier tests for each branching event." However, only the standard multilocus HKA test is mentioned in the methods. The HKA test was designed for two species, and Hey's hka program accepts data from two species. I assume that one of the ingroup species in the HKA outlier test is A. fasciatus, and the other is A. socius, and that the outgroup is A. sp. nov. Texas? Please explain your usage of HKA outlier tests in the methods.
Which species (A. fasciatus or A. socius) was used for the intraspecific polymorphism data? Please include this information in the methods.

Lines 208-214: Please explain how the GA branch determines whether omega values on particular branches are elevated with respect to omega values on other branches.

Validity of the findings

Lines 247-248: Omega statistics cannot differentiate between positive/directional selection and balancing selection when omega is > 1 (or > 0.5, if you use a 0.5 cutoff). Therefore, omega > 1 or > 0.5 is also consistent with balancing selection.

Lines 253-255: Please include the 95% HPD intervals for omega for each gene, in Results and in Table 2. The mean omega values for AK and APBP are greater than 0.5, but do the 95% HPD intervals for the AK and APBP omega values overlap with the 95% HPD intervals for ACG69?

Please conduct pairwise comparisons of omega values between control and candidate speciation genes. This would be helpful in determining which conclusions to put forth.

Lines 271-272: The authors write, "The GA Branch method detected elevated ω classes in all genes except EJAC-SP and SPAG6 (Table 3)." Would the authors please explain this further?
Elevated with respect to mutational steps on the tree between haplotypes of the same species?

Lines 271-277: What conclusions are drawn from these results? The only conclusion from omega statistics I could find in the discussion is in Line 337: "We failed to detect positive selection based on ω."

Lines 278-280: Please explain why you concluded that "only AK and APBP showed advanced lineage sorting for all three species." What qualifies as "advanced"? In Table 4, what does "Pperm" stand for?

Lines 290-293: "However recent divergence can hinder the application of many metrics of molecular evolution that rely on sequence variation since not enough evolutionary time has passed to allow for differences to accumulate between incipient species." Please include references for this statement.

Lines 316-325: Please explain how the argument of Rocha et al. applies to the current manuscript. The authors of the manuscript under review concluded that they did not find positive selection based on omega.

Lines 326-336: I understand the argument in this paragraph, but the authors quote Nielsen as saying, "Therefore in order to detect adaptive evolution due to positive selection, applying
combinations of metrics including ω, Tajima’s or Fu and Li’s D and site frequency spectra, as
well as estimates of population size and recombination rates seems necessary."

How does this recommendation relate to the current manuscript? Estimates of population size and recombination rate were not taken into consideration here, and statistics using SFS did not detect selection. The authors should connect the argument in Lines 326-336 back to their data and conclusions.

Line 337: Authors state, "We failed to detect positive selection based on ω.." I don't entirely understand this conclusion, given than AK and APBP had omega > 0.5 for A. fasciatus vs. A. socius/A. sp. nov. Tex. Pairwise comparisons mentioned above for Lines 253-255 would allow the authors to determine whether AK and APBP have omega values that are statistically significantly higher than genes without species-specific proteomic profiles.

Lines 337: Authors state, "…estimates of D for all genes compared here were not significantly different from neutral expectations." I would like some clarification on the process employed to generate data in Table 1 before I concur that mutations were distributed on genealogies in a pattern consistent with neutral evolution and demographic equilibrium.

Lines 381-401: Two mechanisms of post-mating, pre-zygotic isolation have been identified and characterized by the authors, as mentioned in the Introduction. The authors should connect the putative speciation genes AK and APBP to these mechanisms in the Discussion. Based on the information provided in the Discussion, it seems unlikely that AK and APBP are involved in induction of oviposition. AK and APBP could be involved in CSP, depending on the processes underlying CSP in this species complex. These connections should be explored.

·

Basic reporting

No Comments

Experimental design

Most of the analyses are carried out using online servers (e.g. HyPhy, genealogicalsorting, Mobyle), but the parameters (even if using default options) are not specified. This limits the reproducibility of the analysis, as servers can change their options with new versions. I suggest to clearly describe the type of analysis, explicitly mention the models used and the parameters.
Specifically, inferring positive selection from dN/dS ratio can be more complex than approximating it to 1 or 0.5. This is the case, for example, when few sites along the gene are undergoing positive selection, but the gene is on average conserved. A “site-model” (for example using PAML) would address this question better. It is possible that the authors did this through HyPhy, but it wasn't clear to me that they implemented a site model in addition to the branch model.

Validity of the findings

In line 81 the authors refer to “multiple lines of evidence” to support their conclusion about the patterns of molecular evolution in two genes being consistent with a putative role in speciation. However, I can only find one line of evidence, i.e. the Genealogical Sorting Index, the other tests were not significant.

Some statements, which are essential for the logic of the research are not supported by any reference nor are they explicitly presented as an opinion of the authors. For example in line 101-103, the authors state that “In all, the A. socius complex represents a system whereby speciation is ongoing with relatively few genes contributing to the postmating, prezygotic reproductive isolation between species”. This appears to me an assumption that could eventually be tested through, for instance, a whole-genome study, but that until now remains speculative. More importantly, lines 353-357 present the argument to rule out incomplete lineage sorting and introgression, “However, speciation genes are more likely to become fixed for species-specific alleles early in the process of speciation and therefore are expected to be relatively exempt from incomplete sorting and subject to reduced introgression”, but without being supported by any reference this argument becomes very weak.

·

Basic reporting

No comments

Experimental design

No comments, seems solid.

Validity of the findings

The findings are solid and may be a replication of a previous study, though the rationale and added value should be more clearly defined.

Additional comments

This manuscript presents analyses of five protein-coding genes found in ejaculates and having species-specific proteome profiles along with two control genes that are also expressed in ejaculates yet don’t show species specific profiles. The analyses expand on work reported in Marshall et al. 2011 and includes haplotype network and gene tree approaches that demonstrate evidence for positive selection that was not provided by more traditional sequence-based analyses.

The Allonemobius system is a classic example of speciation via post-copulatory sexual selection and one of the few clear cases where post-mating pre-zygotic processes are the only source of reproductive isolation. The analyses and results presented here seem solid, though I am not an expert in these types of molecular evolutionary approaches. However, the authors need to be very explicit about what additional the data here contribute above and beyond what was presented in Marshall et al. 2011. It seems that the main thrust of the manuscript is that evidence for positive selection is lacking when using sequence-based estimates of ω, yet evidence for positive selection was found for AK and APBP in Marshall et al. 2011. Moreover, the 2011 paper also published haplotype networks for the five of the seven genes examined in this paper. While the current manuscript included A. sp. nov. Tex, it is unclear what these additional data add to the story.

On a more minor note, the Discussion would benefit from a paragraph or two placing these results in the context of the conspecific sperm precedence known to reproductive isolate these species – what might AK and APBP be contributing to that process. Furthermore, SPI was chosen as a control gene, yet some of the patterns it displayed were more intermediate to the putative speciation genes, such as Fig. 2. This result should be discussed as well.

Many aspects of the methods were glossed over – I don’t know what ω classes are and how to interpret them – while other methods were described in almost unnecessary detail (such as dN/dS and ω). It is not clear from Fig. 1 and 2 how only AK and APBP showed advanced lineage sorting, when in Fig. 1, GOT also has fairly clear patterns of haplotype divergence among the three species, and in Fig. 2, all genes except ACG69 have Fisher’s exact test p-values ≤ 0.001. Perhaps I’m missing something here.

The results would benefit from more synthesis and interpretation rather than simply reporting statistics for each lineage. The Discussion often reads like a compilation of what we know about molecular evolutionary statistics and approaches without clearly tying in the relevance of this information to the results obtained, particularly the first four paragraphs.

Lines 383-384: Given AK and APBP are sperm maturation and capacitation proteins, I don't see how interaction with the female reproductive tract would be involved unless sperm mature once inside the female.

---

## Round 0.2 · accepted · Accept

Thank you for carefully revising the manuscript along the lines suggested by the reviewers. Please consider the minor editorial suggestions by reviewer 4 at the proof stage.

Reviewer 1 ·

Basic reporting

None

Experimental design

The authors have addressed detailed comments regarding the methods and the presentation of their results. In my opinion the methods are solid, and sufficient for the paper’s conclusions.

Validity of the findings

None

Additional comments

I liked the additional details regarding the potential mechanisms of the candidate genes in CSP. I think these changes improved the Discussion.

When reading through the revised manuscript I noticed a few typos:

Line 232: there is a typo in “synonymous”

Line 365: there is a typo in “amino”

Reviewer 2 ·

Basic reporting

No comment

Experimental design

The authors have addressed all of my comments from the previous round of reviews.

Validity of the findings

The authors have addressed all of my comments from the previous round of reviews.

Additional comments

I appreciate the effort expended by the authors to address the reviewers' comments. Major analyses were redone, inappropriate analyses were removed, and significant portions of the manuscript were re-written. In my opinion, the manuscript is substantially improved.

·

Basic reporting

The writing in the Introduction is not stellar. Sentences and phrases are often awkwardly worded, which diminishes the clarity of the paper. Sentences are also sometimes too long; a good rule of thumb is to limit sentences to no more than three lines. In some cases, an extra clause is tacked onto the end, which makes the sentence both too long and rather awkward. One example is found in lines 49-52.

Commas should be included more often, such as preceding “including…” (e.g., lines 71 and 76) or a clause beginning with “though…” (e.g., line 80).

The writing in the rest of the manuscript is much better.

Experimental design

This experiment is an improved analysis with better sampling of a previously published dataset, with largely confirmation of old results and some new results. It needs to be stated more clearly (still) how this approach resolves a shortcoming of the previous paper (see comment below). Methods are described in sufficient detail to replicate. Analytical approaches seem to be improved, but again, I lack the expertise to critically evaluate them.

Validity of the findings

Data are robust and statistically sound, and conclusions are well-stated.

Additional comments

The suggested edits below should be interpreted as representative of potential changes that will improve the manuscript rather than being all-inclusive.
1. Insert “that” after “found” on line 26.
2. Lines 25-28: This sentence is generally awkward - shorten and incorporate "lineage sorting of gene trees and allele sharing in haplotype networks" more effectively into the sentence. Alternatively, split this into two sentences.
3. Line 30: Reword “both species branching events of the species tree” to be more clear.
4. Line 36: Change “were” to “was”.
5. Line 37: Insert “which are” or “which represent” after “(APBP),”
6. Line 43: Revise to “induce oviposition in females.”
7. Line 49: Insert a comma after “Therefore”.
8. Line 84: Write out Allonemobius at first mention.
9. Line 90: Reword “organismal” to “study”.
10. Line 114: Insert a comma after “species”.
11. Line 117: More clearly explain here that expanded population sampling is a solution to address these biases.
12. The introduction has been improved, but it should still more clearly delineate what was accomplished in Marshall et al. 2011, that results may have been biased and why, and how you are addressing these shortcomings in this paper. There are bits and pieces here, but explicit language is missing. It was explained more clearly in the rebuttal document than in the text of the manuscript.
13. Line 133: Don’t begin sentences with a genus abbreviation.
14. Line 158: Spell out “first-strand”.